# Food Safety Awareness, Changes in Food Purchasing Behaviour and Attitudes towards Food Waste during COVID-19 in Türkiye

**DOI:** 10.3390/foods12244396

**Published:** 2023-12-06

**Authors:** Irfan Erol, Begüm Mutuş, Naim Deniz Ayaz, Julian D. Stowell, Belgin Sırıken

**Affiliations:** 1Faculty of Health Sciences, Lokman Hekim University, 06510 Çankaya, Türkiye; 2Faculty of Health Sciences, Atılım University, 06830 İncek, Türkiye; 3Sabri Ülker Food Research Foundation, Üsküdar, 34692 Istanbul, Türkiye; begum.mutus@yildizholding.com.tr (B.M.); julian.stowell@btinternet.com (J.D.S.); 4Department of Food Hygiene and Technology, Faculty of Veterinary Medicine, Kırıkkale University, 71450 Yahsihan, Türkiye; naimdenizayaz@kku.edu.tr; 5Department of Aquatic Animal Diseases, Faculty of Veterinary Medicine, Ondokuz Mayıs University, 55139 Atakum, Türkiye; bsiriken@omu.edu.tr

**Keywords:** consumer behaviour, COVID-19, food hygiene, food waste

## Abstract

(1) Background: The COVID-19 pandemic brought the key issues of food security, food safety, and food waste into sharp focus. Türkiye is in the enviable position of being among the top ten agricultural economies worldwide, with a wide diversity of food production. This survey was undertaken in order to gain insights into consumer behaviour and attitudes in Türkiye with respect to these issues. The objective was to highlight strengths and weaknesses, identify areas for improvement, and present strategies for the future. (2) Methods: This survey was carried out between April and May 2022 in 12 provinces throughout Türkiye. Face-to-face interviews were performed with 2400 participants representing a cross-section of ages, educational attainment, and socio-economic categories. The findings were evaluated statistically. (3) Results: The results provide an insight into attitudes and behaviours, both pre-COVID-19 and during the pandemic. In several ways, the pandemic enhanced knowledge and improved behaviour, leading to improvements in diet and reductions in food waste. However, worrying concerns about food safety persist. Specific attention has been given to understanding patterns of bread consumption, particularly in consideration of waste. (4) Conclusions: It is hoped that the results of this survey will increase dialogue between the components of the food sector, encourage education initiatives, and contribute to improving food safety and security and reducing food waste in Türkiye and beyond.

## 1. Introduction

Food security and sustainability are issues of global importance [1]. Seen against a backdrop of the global financial crisis of 2008, a rapid increase in world population, global warming, natural disasters, and, recently, the COVID-19 pandemic, it is vital that consumers are empowered to take individual responsibility to secure a sustainable future for our planet. Türkiye ranks among the top ten global agricultural economies [2] and is exceptionally well-placed to grow a wide variety of food crops. However, despite this, nutrient deficiencies and inequalities persist against a backdrop of vast amounts of food waste [3].

Foods may pose a public health risk if contaminated with physical, chemical, and/or microbial hazards from farm to fork, including production, processing, distribution, and storage [4]. According to WHO data, 600 million people get sick, 420,000 people die, and 33 million healthy life years (DALYs) are lost every year due to food-borne diseases worldwide [5]. Food-borne infections and intoxications are generally caused by foods of animal origin and fresh fruits and vegetables grown using contaminated water or inadequately composted manure [5,6]. Although reports on food-borne disease cases and outbreaks are published mostly in developed countries, it is known that these outbreaks are more common in developing societies due to a lack of food safety infrastructure [7,8,9]. Factors such as globalisation, migration to cities, and shifting demographics due to refugees, as well as radical advances in food production technology, increasing incomes, and altered purchasing preferences, all contribute to changing food consumption habits around the world, and food safety concerns are evolving concomitantly [6].

Studies have highlighted that factors including cost and practical preparation are seen as more important than food safety risks, especially by low- and middle-income consumers [10]. In addition, poor hygiene concerns outweigh the nutritional content of foods in determining purchasing preferences [11,12]. These considerations may lead consumers to a diet that is cheap, accessible, and easy to prepare but that is insufficient in essential amino acids, vitamins, and minerals, as well as suboptimal in terms of salt, added sugar, and trans-fat content [13,14].

Although home-made food is considered a safer and healthier option by consumers in some studies [12], fast food and packaged foods are increasing in popularity due to their accessibility and affordability, especially in urban life [15,16,17]. However, inadequate food safety standards and compliance can increase the risk of food-borne diseases and epidemics, especially in developing countries [4,16].

Another important issue is food security. According to the State of Food Security and Nutrition in the World (SOFI) report, it was stated that 691 to 783 million people faced hunger in 2022. It is quite striking that the report shows that hunger increased by 122 million people compared to 2019. The report stated that this increase was caused by the COVID-19 epidemic, repeated weather shocks, and wars around the world [18], especially in Ukraine [19].

Related to food security, food waste is a growing global problem with negative effects on societies, economies, and the environment [20]. The FAO’s 2019 report calculated that approximately 14% of the world’s food supply was wasted, and the economic value of this was USD 400 billion [21]. The FAO’s 2022 report states that if the food currently wasted annually could be provided to consumers, approximately 1.26 billion hungry people could be fed [22]. Food waste also negatively affects the environment via greenhouse gas emissions; it is responsible for 8% of total greenhouse gases [20].

Food wastage occurs as a result of bad food management practices at all stages in the food chain, including planning, purchasing, storage, cooking, eating, and subsequent leftovers [23]. Recent studies have shown that the COVID-19 pandemic was responsible for significantly changing food supply transport and usage patterns [24,25]. These changes include buying more food than needed and more regular and frequent shopping. Both reduced and increased food wastage has been observed depending on individual situations [24,26].

Globally, the 2008 financial crisis adversely impacted consumer behaviour by causing a contraction in production, resulting in increased food prices and a corresponding decrease in purchasing power. Subsequently, the situation improved until the COVID-19 pandemic, which occurred from 2020 onward [27]. The vital importance of maintaining a healthy and sustainable food supply was then brought into sharp focus. Measures taken to minimise the impact of the pandemic and consequent changes in consumer behaviour also necessitated changes in food business models and the food supply chain. The strategic importance of food security and safety achieved greater prominence. In this context, in order for food industry stakeholders to become more resilient to crises and create a vision for the future, it is necessary to have accurate information on the main issues affecting consumers’ purchasing behaviour. However, in the literature review, it is noted that there are no comprehensive studies across the country in Türkiye, which is an important producer and market in Europe.

The purpose of this consumer survey is to focus on Türkiye, gaining insights into consumers’ attitudes toward food safety, security, food waste, and purchasing behaviour before and during the COVID-19 pandemic. It is hoped that the results obtained will help to inform stakeholders and highlight priorities for future initiatives [28].

## 2. Materials and Methods

### 2.1. Profile of Participants and Survey Design

This survey was carried out during April and May 2022 in 12 provinces representing the geographic regions of Türkiye to ensure that the survey is representative of the whole country, taking into account different cultures and food habits. Face-to-face interviews were conducted with 2400 individuals. The subjects participated in the survey voluntarily. A random sampling method was used to recruit participants, and the number in each region was based on the population in that region (Figure 1). All respondents were aged 18 and over and were responsible for food purchasing in their household. The sample size was calculated according to the rule of taking 10 times the number of scale items, and the survey was carried out with a margin of error of ±2 and a 95% confidence interval.

Within the scope of this survey conducted in the centers of the most populous provinces from the 12 NUTS-1 (Statistical Territorial Units Classification/NUTS) level regions determined by the Turkish Statistical Institute (TUIK), interviews were held with individuals who represent the household, could provide information about the entire household, and are responsible for household consumption. According to the NUTS classification system, there are 12 NUTS-1 regions, 24 NUTS-2 sub-regions, and 81 NUTS-3 provinces in Türkiye. This classification was made according to population size, taking into account social, economic, and geographical factors [29].

According to the 2022 results of the Turkish Address-Based Population Registration System (ADNKS), the population of Türkiye was 85,279,553 as of 31 December 2022. Of this number, 50.1 percent (42,704,112) were men and 49.9 percent (42,575,441) were women. Among this population, 62,701,155 people were adults [30]. The sample size, which was determined as at least 2400 to represent approximately 19,000,000 households in Türkiye with a margin of error of 2%, was distributed to 12 large cities in proportion to their populations. A total of 8400 individuals live in the visited households. Therefore, the average household size was determined as 3.5 individuals. This value is also compatible with the results of the 2011 Türkiye Family Structure Survey.

The selected cities comprise the provinces with the largest population center within the NUTS-1 level region. The research was designed with the assumption that the largest province selected within each NUTS-1 region represents the entire region. The selection of addresses within the province, which determined the households to be interviewed, was conducted randomly. Within the scope of the field study, an adult member of the household (preferably the person who brings the most income to the household or his/her spouse) was interviewed in a total of 2400 households. If there was more than one person responsible for food shopping in the household, the person with the closest birthday among the responsible persons was included in the research to ensure randomness within the household. Therefore, purposeful sampling was applied in the selection of the sample within the household, and a simple random sampling method was applied in case there was more than one suitable individual within the household.

Before the field study of the research, a plot study was carried out on 50 households to test the functionality and understandability of the questionnaire. According to the plot study’s findings, the revised forms were finalised and made ready for application, and the study was carried out with the face-to-face interview method based on the questionnaire. During the plot study, it was observed that the interviews lasted approximately 20 min. After the fieldwork, 30% of the interviews conducted by each interviewer were checked by the back check method by contacting the households again.

### 2.2. Key Objectives of the Survey

In order to gain insights into food purchasing habits, food safety, and attitudes toward food waste before and during the COVID-19 outbreak, a questionnaire containing the following four parts was prepared.

Part 1 involved the collection of demographic data, including age, gender, marital status, household income, socio-economic status, and education level of the participants. According to the household income and education level, the socio-economic status of the participants was classified as follows:

A: The person who brings income to the household is a manager or business owner and has a high level of education (university and/or higher degree);

B: The person who brings income to the household is a manager, business owner, or a qualified expert and has a medium–high level of education (high school or university);

C1: The person who brings income to the household is a housewife (with additional income) or a worker and has a high level of education or has a workplace and a low–medium level of education (primary or high school);

C2: The person who brings income to the household is a housewife, student (with additional income), or worker and has a low–medium level of education (primary or high school);

DE: The person who brings income to the household is a housewife, unemployed (with additional income), or worker and has a low level of education (primary school).

Part 2 set out to determine the knowledge level of food safety and food safety concerns. The following questions were devised and asked of the participants in order to address this aspect:The relative importance of the different elements involved in the purchase of food products;The knowledge level of the concept of food safety;The perception of food safety;Awareness of information resources available relating to food safety;Food safety status;Reactions to negative situations due to unsafe food consumption;The level of recognition of food safety systems;Perspectives on food control;Attitudes toward food hygiene;Actions where a lack of hygiene is observed in the production and sales of foods;Knowledge of the official organisation responsible for food safety;Follow-up on the fraudulent products or companies’ disclosure list;Willingness to pay more to facilitate better food inspection;Understanding of health issues related to food consumption;Use of the 174 food line resource;Knowledge of the health aspects of foods;Encounters with spoiled and faulty foods;Occasions of notifying about broken and faulty foods;Reasons for not reporting defective and faulty product complaints;The status of receiving feedback on complaint notifications;Current attitudes and behaviours with regard to food waste;The most common reasons for wasting food;Information on bread consumption and bread waste.

Part 3 focused on determining the effects of the COVID-19 pandemic on food purchasing behaviour and attitudes. The questions addressed were:Food purchasing behaviours before and during COVID-19;The main sources of food purchases and food supply routes;Data regarding food stocks maintained at home during the pandemic compared to those in the pre-COVID-19 period.

Part 4 sought to determine the knowledge and attitudes of consumers with regard to food waste. Another objective of this part was to determine behaviours specifically relating to bread waste. Questions included:How decisions are made regarding whether food is spoiled;What happens to unused foods;Steps taken to minimise food waste;Reasons for wasting food;Attitudes about wasting bread;How the source of bread correlates with wastage;Types and quantities of bread consumed;Influence of source, packaging, and freshness on bread wastage.

The data obtained in the survey were processed together with the demographic data, and their statistical significance was determined.

### 2.3. Statistical Analysis

Statistical techniques were applied to identify differences between the situation prior to and during the COVID-19 pandemic [31]. Pearson correlation was applied to evaluate the differences between food purchasing habits and to measure the relative importance of the various factors involved in food purchasing before and during the pandemic. The Kruskall–Wallis test was applied to highlight differences in food safety and consumer attitudes toward food safety and to evaluate whether foods are safer than in previous years. The Shapiro–Wilk test was conducted to examine current food safety aspects in comparison with those in previous years. Tukey’s post-hoc test was used to further examine the differences highlighted during the analyses. A chi-square analysis was carried out to evaluate differences in food safety and food stocks before and during the COVID-19 pandemic. A multiple regression model was used to analyse the data on food waste.

## 3. Results

### 3.1. Demographic Characteristics of the Participants

Half the participants in the survey were women, and among them, 66.0% were married. The average age of all participants was 39, and they were evenly distributed among age groups (Figure 2).

The distribution of participants among socio-economic groups was A-B (14.2%), C1-C2 (64.9%), and D-E (20.9%). Their education attainment level was declared as primary school (41.4%), high school (40.8%), and university and/or higher degree (17.8%).

### 3.2. Consumers’ Knowledge of Food Safety and Food Safety Concerns

Figure 3 summarises the participants’ responses regarding the importance of various factors in food purchases. It can be seen that “expiry date”, “label information”, and “quality and taste” headed the list.

Data obtained on changing attitudes towards food purchases are summarised in Figure 4. When participants compared their current thinking with that in the past, some 60.8% considered that prices were better. Surprisingly, only 19.5% of respondents found the label information to be better now than in the past. Attitudes towards packaging remained unchanged for 44.6% of participants.

With regard to food safety, 33.9% of participants stated that they have heard of the concept, whilst 66.1% have not. A greater proportion of participants were aware of the concept, 37.1% versus 30.8% respectively. The 18 to 24 age group (41.2%) was most aware, and regarding socio-economic status, awareness was as follows: A-B: 39.0%, C1-C2: 43.4%, and D-E: 28.9%.

Participants were asked to identify the statement of food safety. Their responses were as follows:Food that is brought to the market after being controlled by legal institutions at every stage, from production to consumption: percentage of respondents (27.9%);To ensure the safety of food against chemical hazards (25.3%);To ensure that consumers have access to adequate food (22.9%);Food without an expiry date (17.2%);Microbially safe food (6.8%).

Information on food safety is derived from multiple sources. Participants who were aware of food safety identified the statement or statements that best described their main source(s) of information as follows:TV and radio programmes (70.1%);Internet or social media (48.5%);Family and close circle of contacts (31.8%);Scientific articles and books (26.4%);Newspapers and magazines (25.9%);Experts (18.6%).

Compared to previous years, 21.9% of the participants thought that food is safer now. The highest proportion of those who declared that food is safer now were in the 40–44 age group (27.7%), whilst the highest proportion of those who thought that it is less safe were in the 35–39 age group (15.9%). With regard to socio-economic status, the C1-C2 group (24.4%) were most positive about food being safer now than in the past.

The survey measured consumers’ reactions to the consequences of their negative experiences due to unsafe food consumption. Respondents identified key points such as:Avoided consuming the unsafe foods for a while (24.1%);Did not do anything or did not take the situation into account (37.1%);Reported the situation to the authorities (11.1%).

In terms of the recognition of food safety systems (Figure 5), 87.8% of respondents recognised the Turkish Standards Institution (TSE) standards as a food safety system, and 62.0% claimed to know the International Organization for Standardization (ISO) standards.

The main reason (62.7%) why participants thought that there was no measurable food control was that sanctions or penalties do not provide a sufficient deterrent. Some 60.3% of the participants considered that hygiene rules are not adequately taken into account in food production. The 18–24 age group (46.6%), high school graduates (40.9%), and individuals of C1-C2 socio-economic status (41.4%) were most convinced that hygiene standards are taken into account, while those over 50 (71.5%), primary school graduates (68.1%) and those of D-E socio-economic status (76.9%) were least convinced.

Participants identified the following areas where food hygiene is lacking:During food processing (57.4%);During raw material production (48.1%);During storage (38.1%);During transportation (37.4%);At ready meal producers, including restaurants and patisseries (22.9%);At wholesale and retail grocers (21.3%).

According to the findings, 39.0% of participants considered the Ministry of Health to be the official institution responsible for food safety, followed by the Ministry of Agriculture and Forestry (29.7%), TSE (19.0%), Municipalities (11.5%), and the Ministry of Industry and Technology (0.8%). In Türkiye, companies engaged in fraudulent production are identified by way of inspections by the Ministry of Agriculture and Forestry. However, it was determined that 62% of the participants were not aware of this.

Participants in this survey were questioned as to whether they would be willing to pay more for food if it had been inspected by an independent organisation to ensure its safety. A total of 67.5% of the participants stated that they would not pay more. The 18–24 age group was most likely to be willing to pay more (40.2%), followed by the 40–44 age group with 38.8%. For other age groups, the level of acceptance was between 26.1% and 31.9%.

Of the participants, 48.8% defined “ALO 174” as the contact telephone number where all kinds of food-related complaints and consumer demands are processed. Some 31.6% of the participants stated that they had not heard of this facility, and 16.5% thought it was a number that consumers could call to learn about the quality of food.

In Türkiye, based on this survey, 45.5%, 32.7%, and 21.8% of consumers thought that the foods consumed are not healthy, partially healthy, and healthy, respectively. The highest proportion of those who considered foods to be safe were individuals between the ages of 40–44 (29.2%) and in the A-B socio-economic category (24.9%). The highest proportion of those who do not consider foods to be safe were individuals aged 50+ (56.5%) and in the D-E socio-economic category (67.5%).

In the survey, 30.0% of the participants stated that they had encountered spoiled and adulterated food products. Additionally, 77.6% of the participants said they would lodge a complaint when encountering broken and faulty food products. Among complainants, 88.9% stated that they had received positive feedback. Reasons given for not reporting a complaint included:Unwillingness to deal with the complaint (62.1%);Not getting results (51.6%);Not knowing to which entity the complaint should be made (14.9%).

Among the health problems associated with food, cancer ranked first with 34.4%, followed by obesity (25.0%), diabetes (19.5%), food poisoning (10.9%), and cardiovascular diseases (9.7%).

### 3.3. Food Purchase Behaviours before and during the COVID-19 Pandemic

In this survey, behavioural patterns toward food purchases were examined before and during COVID-19. During the pandemic period, the purchasing frequency of fresh vegetables, fruits, fresh meat, chicken, fish, and legumes increased compared to pre-COVID-19 (*p* < 0.000) (Figure 6).

Figure 7 summarises the main sources of food purchases as determined by the survey. Supermarkets constituted the main source of food purchases both before and during the COVID-19 period. During COVID-19, there was an increase in food purchased from street vendors and home delivery–online channels compared to the period before COVID-19. A slight decrease in food supply via takeaway (42.7% to 41.1%) and ready-to-heat/cook meals (42.3% to 33.7%) was seen during the COVID-19 pandemic, while the rate of food preparation at home (64.9% to 65.9%) increased slightly.

The survey assessed food supplies in homes before and during COVID-19. A total of 55.1% of participants reported that food stocks remained the same, 28.2% said that stocks increased, and 16.8% reported lower stock during COVID-19 than before. Higher stocks during COVID-19 were reported according to socio-economic status as A-B (22.6%), C1-C2 (24.6%), and D-E (43.2%). Chi-square analysis was carried out to evaluate differences in perception of food safety and differences in food storage trends at home in the periods before and during COVID-19. A decrease in the perception of food safety was inversely related to food stocks maintained at home (*p* = 0.000).

### 3.4. Knowledge of Food Waste: Current Attitudes and Behaviours to Minimise Waste

The focus of the survey on food waste considered a number of parameters, including detection of spoilage, unconsumed food management, methods for minimising food waste, and the main reasons for wasting food.

In order to determine whether food was spoiled, 70% of consumers smelled the food, 58.9% carried out a visual inspection, and 34.9% tasted it.

Regarding unconsumed food, respondents stated:“I give it to stray animals” (56.2%);“I use it in a different meal” (55.1%);“I keep it in the deep freezer” (34.1%);“I distribute it to those around me” (19.0%);“I throw it away” (11.8%).

When measures taken to minimise household food waste were evaluated, the statement “buying as much as we need” in order to minimise food waste was the most popular, with 65.6% of respondents highlighting this option. Controlling the food stock at home and making a shopping list were preferred by 49.0% of the participants, followed by 33.3% of respondents whose primary objective was to use foods left over at the next meal, and 14.8% of respondents preferred to keep leftover foods in a suitable condition for future use.

Among the most common reasons for wasting food, 26.1% of the participants stated that this was due to the food having passed its expiry date, 24.2% had bought more food than needed, 24.1% had food which had become mouldy, spoiled in some other way, or lost its freshness, 18.9% had failed to use appropriate storage conditions, and 6.7% of respondents were living with others but had failed to communicate appropriately regarding what foods were needed, leading to excess purchases.

### 3.5. The Example of Bread Waste

A particular focus of this section of the survey was to evaluate current attitudes and behaviours with respect to bread waste. A total of 84.4% of participants stated that they purchased bread outside the home, 4.4% stated that they made it at home, and 11.3% stated that they did not consume any bread. The largest percentage of those who purchase bread outside the home (91.9%) are in the 25–29 age group, while the largest percentage of responders who make bread at home are in the 30–34 age group (8.4%). Some 81.6% of respondents who stated that they purchased bread outside the home preferred unpackaged bread, specifically regular/white bread in 200 g lots. The highest proportion of those who purchased packaged bread were in the 25–29 age group (27.1%), university graduates (27.2%), and A-B socio-economic status (3.7%), while the highest proportion of those who bought unpackaged bread were individuals aged 50 over (88.0%), primary school graduates (83.8%), and from the D-E socio-economic category (90.9%).

In the survey, 70.6% of participants stated that they consumed regular/white bread, followed by bread from whole wheat (10.9%), bran (6.2%), multigrain (3.9%), rye (3.0%), sourdough (2.0%), and buckwheat (1.9%). It was determined that an average of 3.6 loaves of bread are purchased daily (per house) in Türkiye. Of these, 1.5 were purchased in the morning, 1.5 in the evening, and 0.6 in the afternoon. With regard to storing leftover bread, 42.4% of the participants preferred to use a bread cabinet, 36.9% kept it in bags, and 20.7% kept it in the refrigerator. Feedback from the questionnaire indicated that 58.1% of participants consumed stale bread, 55.8% used it in different meals or desserts, 33.0% kept it frozen, 31.7% gave it to animals, 27.6% heated it in the oven, and 7.4% stated that they throw it away. A total of 75.6% of participants stated that no bread was wasted at home, and 60.6% stated that no bread is left on the table after a meal. The participants were of the opinion that most bread wastage is in restaurants, hotels, and other public eating places (39.8%). Bread consumption and wastage are summarised in Figure 8.

In the survey, 60.1% of participants considered it important to raise awareness about food waste in basic education. The highest proportion of those advocating raising awareness about food waste were in the 18–24 age group (68.9%), university graduates (70.3%), and A-B socio-economic status (65.5%). Conversely, the highest proportion of those who do not see it as necessary were over 50 years old (45.2%), primary school graduates (47.1%), and from the D-E socio-economic status (45.1%).

## 4. Discussion

In this survey, only 21.9% of participants considered that the food in Türkiye is safer today than in previous years. This is in stark contrast to the situation in the UK, where 92% of the participants in a recent survey by the Food Standards Agency (FSA) considered that the food they purchase is safe [32]. Baker et al. [33] highlighted the concerns of low-income consumers globally regarding food safety. This ties in with the findings of the current survey whereby the highest proportion of those considering food to be safe were in the A-B socio-economic group, while those most likely to consider food unsafe were in the D-E socio-economic group. Pradeilles et al. [34] identified the high risk of poor hygienic conditions, environmental sanitation, food contamination, and food adulteration for low- and middle-income groups living in cities. This concurs with the findings in this survey.

Participants in this survey reported that television programmes were 3.8 times more effective as a source of food safety information compared to information from experts in the field. Studies conducted in several countries identified common sources of food safety information as university education, family and friends, family physicians, journal articles, social media, educational environments, governmental institutions, cooking classes at school, and television programmes, among others [35,36,37]. In a study conducted in Sweden, the leading source of food safety information (45%) was identified as family and friends. Some 21.1% of participants reported that they had received food safety training, and 35.6% stated that they had experienced a course in food hygiene/safety and/or microbiology [38]. In Sweden, training about hygiene and cleanliness while processing, preparing, and storing food is compulsory in primary schools [38]. A principal source of food safety information is family communication [36].

In our survey, the expiry date was determined to be the primary consideration when purchasing food products (77.3%), followed by label information, quality, and taste criteria. In their study, Armstrong et al. [32] found that 80% of participants reported that they frequently check the expiry date of foods while shopping.

In our survey, the highest proportion of participants who purchased packaged bread were in the 25–29 age group, university graduates, and individuals from the A-B socio-economic group. Conversely, the highest proportion purchasing unpackaged bread was in the 50+ age group, primary school graduates, and from the D-E socio-economic group. This is in contrast to the findings of Baker et al. [33], who determined that low-income consumers tend to avoid fresh foods due to safety concerns, preferring packaged and processed foods. In Türkiye, the reason why consumers in low-income groups tend to prefer unpackaged bread is most likely due to the lower product cost rather than food hygiene.

Although the Ministry of Agriculture and Forestry is the official authority responsible for food controls in Türkiye, only 29.7% of those who participated in the survey gave the correct answer. In a study conducted in the United Kingdom, it was determined that 91% of the participants knew about the Food Standards Agency (FSA), 75% trusted the FSA to ensure food safety, and 91% were confident that the FSA would take appropriate action if a food-related risk were detected [32].

In this survey, only 19.5% of participants stated that food label information is better today than in the past. In the FSA study referred to above, 86% of the participants stated that they trust the information on food labels to be correct [32]. Regarding consumers’ perceptions of food production hygiene standards in Türkiye, 60.3% of participants in our survey considered that hygiene rules are not adhered to in food production. However, in the FSA study, 78% of respondents reported that they trust the food supply chain. It was determined that consumers in England, Wales, and Northern Ireland trust farmers (90%), stores, and supermarkets (86%) more than takeaways (51%) and food delivery services (39%) [32].

Our survey determined that, in Türkiye, most food shopping occurred in supermarkets both before and during the COVID-19 period. In addition, during the pandemic, purchases from street vendors and home delivery–online shopping increased compared to the situation before. Similarly, supermarkets were confirmed to be the preferred food shopping sources in the UK (85%) [32].

In our survey, it was determined that the frequency of purchasing fresh meat, chicken, fish, and legumes increased during COVID-19 compared to before, and the frequency of consumers preparing food at home also increased. Similar results were obtained in other studies, both in England, Wales, and Northern Ireland [32] and in the USA [39]. In the USA, increased availability of time, the belief that eating out was riskier, and a decrease in income were cited as reasons for the increase in home consumption during the COVID-19 period. In this survey, it was also stated that increased home storage of food may negatively affect the robustness of the food supply chain [39].

Studies have shown that home consumption of food is especially prevalent in low- and middle-income groups. In addition to the economics, consumers’ food safety concerns can reduce the rate of eating out. Although foods prepared and cooked at home were considered safer, healthier, and more nutritious by consumers, street foods can be preferred because they are easily accessible and affordable. These advantages can outweigh concerns about food safety in consumers’ choices [8].

Measuring willingness to pay is considered an important tool in the understanding of consumers’ attitudes and views toward food sustainability [26]. Price elasticity is a key parameter. In our survey, 32.5% of participants stated that they would pay more for food that was inspected by an independent organisation to ensure safety.

Food purchasing behaviour has changed in response to the stress and uncertainty caused by COVID-19. Panic buying is a feature that has led to increased food waste [38]. In Türkiye, 55.1% of participants in our survey stated that the food stock in their homes remained almost the same as pre-COVID-19. However, 28.2% of participants stated that their food stocks increased.

Many socio-demographic changes during the COVID-19 pandemic impacted food waste. These include household size, loss of income, restrictions, increased cooking skills, better meal planning, more efficient storage, and reduced eating out in restaurants, as well as factors such as stress, depression, fear, and anxiety [31]. These food waste patterns vary from country to country as well as from individual to individual [24,40,41,42]. Decreases in domestic food waste during the COVID-19 pandemic were reported in countries including Italy, Romania, Portugal, and the USA [19,24,25,43,44]. Conversely, studies have reported increases in food waste in Serbia and Thailand [23,45]. In our survey, multiple reasons were given for decreasing food waste, but it was not possible to quantify the overall effect. One study determined that the amount of food purchased per shopping visit increased, reducing the number of visits required and, hence, hopefully, the risk of infection [46]. Similarly, in a UK study, it was determined that consumers prepared shopping lists, decreased their shopping frequency, and purchased more food per visit. The study predicted that this changing behaviour would lead to a 34% reduction in the amount of discarded uneaten food compared to a similar study conducted in November 2019 [47]. Consumers in Italy reported that they wasted less food during the COVID-19 period. In this study, it was determined that the highest decrease in food waste occurred among young people [44].

Reusing leftover food is considered to be one of the most effective ways to prevent or reduce food waste [43]. In this survey, 33.3% of participants reported that they reduced food waste by reusing leftover food promptly during the COVID-19 pandemic. Concerning bread, 55.8% of participants reported that they reused bread in different dishes and desserts, and 27.6% reported that they reheated bread in the oven.

Studies have shown that there is a significant relationship between food waste and age, education, and socio-economic status [42,48]. Generally, it was ascertained that young people and those who work create more food waste than older people and people who do not work and/or are retired [42]. Those with higher education degrees were found to be more concerned with sustainability and reducing food waste [48].

## 5. Conclusions

A key finding of our survey is that awareness of the critical issues of food safety and food waste increased during the COVID-19 pandemic. This is positive. However, it is not clear that lessons learned have been sustained subsequent to COVID-19, and in fact, the converse may be true. There is still distrust of food safety and the role of the relevant authorities in Türkiye compared to other countries where data exist. Socio-economic status, age, and household income all have an impact on attitudes about these key issues. However, above all, it is education, or rather lack of it, that determines current practices. Best practices will only be achieved with comprehensive education beginning in the home and primary school and continuing throughout life. Reliable and trusted sources of information are essential, and only in this way can all stakeholders become empowered to make a positive contribution to an improved future for society. It is hoped that the findings from this survey can help to inform future strategies and reduce the negative impact of pandemics that may arise in the future. Moreover, this survey was conducted during the pandemic period, and it is possible that those conditions affected consumer behaviour periodically. For this reason, it is important to repeat such studies at regular intervals so that the food industry can create future visions.

## Figures and Tables

**Figure 1 foods-12-04396-f001:**
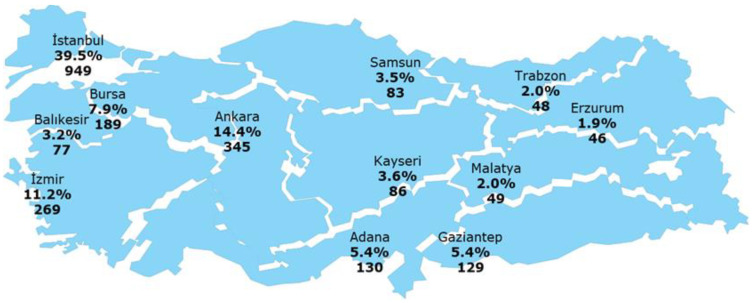
Distribution of the participants in the survey by provinces.

**Figure 2 foods-12-04396-f002:**
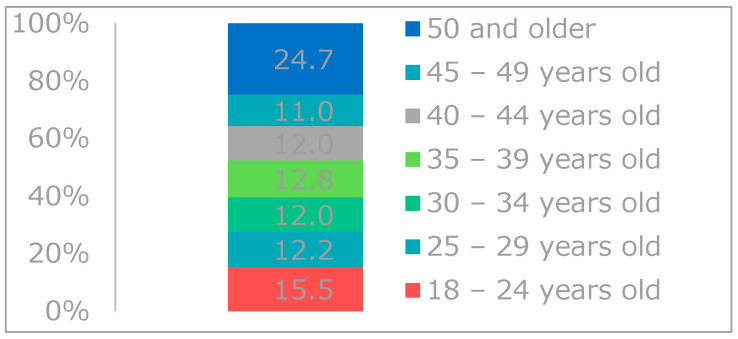
Age of the participants in the survey.

**Figure 3 foods-12-04396-f003:**
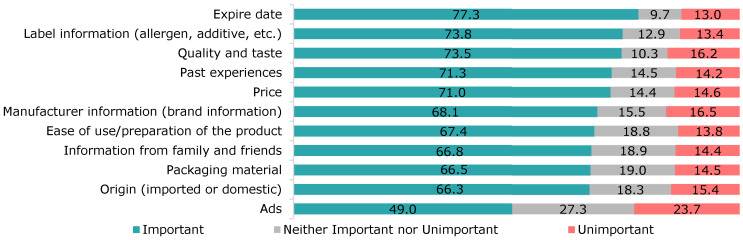
Importance of various factors in the purchase of food products.

**Figure 4 foods-12-04396-f004:**
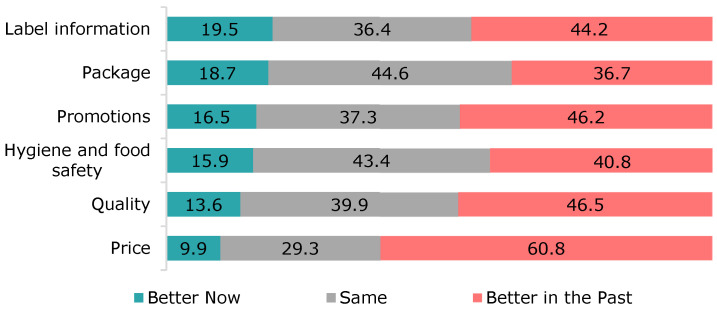
Changes in attitudes towards food purchases.

**Figure 5 foods-12-04396-f005:**
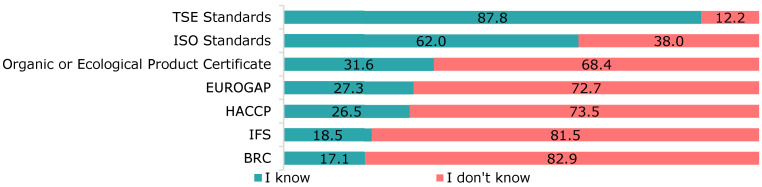
Recognition of food safety systems.

**Figure 6 foods-12-04396-f006:**
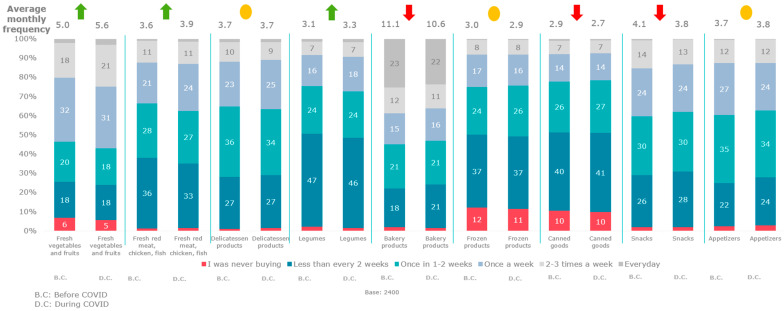
Purchase frequency of food types before and during COVID-19.

**Figure 7 foods-12-04396-f007:**
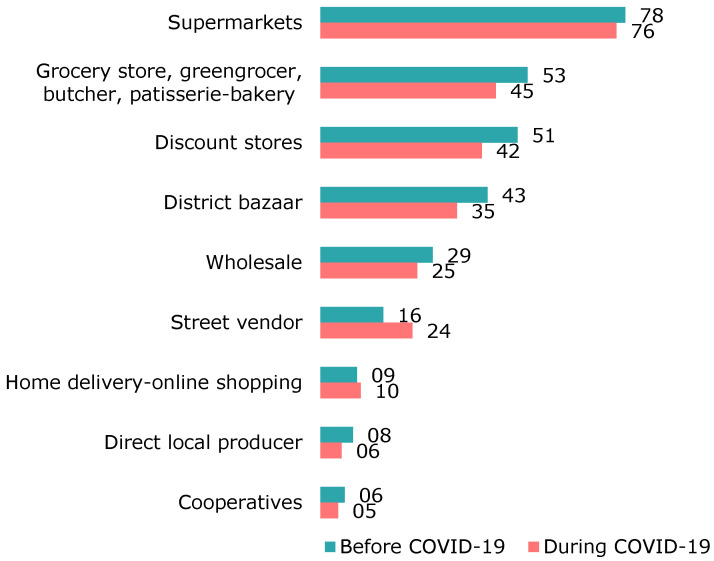
Main sources of food purchases before and during COVID-19.

**Figure 8 foods-12-04396-f008:**
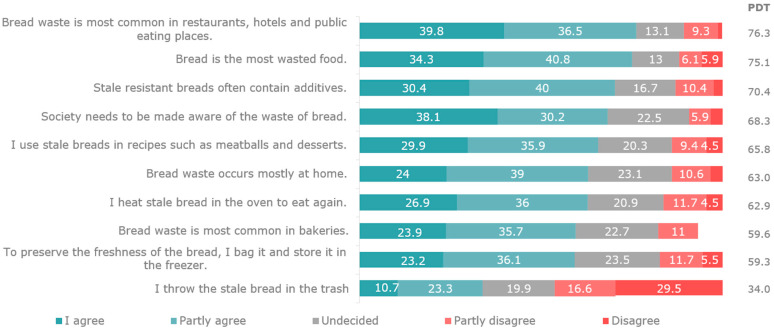
Evaluation of bread consumption and wastage.

## Data Availability

Data are contained within the article.

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
