# Peer review of "Food Safety Awareness, Changes in Food Purchasing Behaviour and Attitudes towards Food Waste during COVID-19 in Türkiye"

_foods, 2023, doi:10.3390/foods12244396_

Round 1

Reviewer 1 Report

Comments and Suggestions for Authors

Irfan Erol et al. submitted to Foods an article focusing to the food safety awareness and the changes in food purchasing behaviour during pandemic in Türkiye.

Below are the aspects that need to be clarified and implemented:

- there are 2400 respondents, but what is the "denominator"? How represented is the level of compliance, compared to the total population? It is absolutely essential to clarify this aspect, to understand whether it is possible to apply statistical inference, otherwise this survey could only be considered a local spot study.

- How were respondents recruited? At the exit of the supermarkets?

- Has the questionnaire been validated in terms of intelligibility, through a pilot phase?

- The English language and typo must be correctly edited.

At present, this manuscript requires clarification regarding the methods used to conduct the survey. Moreover, the English language needs to be proofread, probably via an English editing service, thank you.

Comments on the Quality of English Language

Extensive editing of English language required.

Author Response

The authors are grateful to reviewer’s for their valuable comments and corrections. We have made revisions to our manuscript based on the feedback provided by the reviewer’s. Our responses are listed below.

The English of the article was reviewed by one of the authors, whose native language is English, and necessary corrections were made.

Reviewer 1

  1. There are 2400 respondents, but what is the "denominator"? How represented is the level of compliance, compared to the total population? It is absolutely essential to clarify this aspect, to understand whether it is possible to apply statistical inference, otherwise this survey could only be considered a local spot study.

Response: Explanation has been added to the relevant section (page 6-7).

  1. How were respondents recruited? At the exit of the supermarkets?

Response: Explanation has been added to the relevant section (page 6-7).

  1. Has the questionnaire been validated in terms of intelligibility, through a pilot phase?

Response: Explanation has been added to the relevant section (page 6-7).

  1. The English language and typo must be correctly edited.

Response: The English of the article was reviewed by one of the authors, whose native language is English, and necessary corrections were made.

Reviewer 2 Report

Comments and Suggestions for Authors

Congratulation to the authors for the in-depth and wide study carried out..

Line

Original version

Suggestions/comments

Line 14

food safety, food security

I should put the first key issue, “food security”

Line 17

To inform

To present?

Line 18

Turkiye is in the enviable position…

I would suggest this sentence to the second due to the fact it is the subject of this study.

Line 27

Inform dialogue

Please clarify

Line 40

From ‘farm to fork’

I would suggest to add more information regarding this frase.

Line 41

Data from WHO

I propose to add also the DALY-disability-adjusted life year

Line 45

This is not a hypothesis. Propose introducing the lack of Food Safety Infrastructure in developing countries and the full-scale operation in developed countries.

Line 47

Shifting demographics

What is the intention of this phrase? Be clearer in explanation.

Line 63

Food waste

After elaboration of Food safety and food waste key factors, less information is considered about food security in the time of crises.

Line 66

Please correct. Avoid double spaces.

Line 69

Released into the environment

Please avoid repeatedness

Line 72

Suggest to consider also the transport

Line 78

Adversely impacted

Please explain

Line 91

Study

Please unify the terminology “study” and ”survey”

Line 92

Geographic regions of Turkey

Except the geography, what do you think about the cultural diversity or something else connected with regions?

Line 111

Please unify the indent

Line 174

A-B; C1-C2, and D-E

Please explain the groups categorization

Line 179

Quality and taste

Please put in quotes this phrase

Line 180

In my opinion the only option stands ‘ads’ as different compared with others.

Line 183

Please merge two sentences

Line 212

Today

Please change this word e.g. actually

Line 219

Said foods

Please explain

Line 240

Outlets

This word is not so common in food sector. Please unify the terminology according to www.fmi.org/food-industry-glossary

Line 360

Since these three divisions belong to the UK, I would suggest the second. Even though is not stated Scotland.

Line 371

Ofof

Please correct

Line 392

In Turkiye

In my opinion, not only. What about the access to the retail actors? There is a difference in urbanization characteristics between Western countries and developing countries. Quick access to the neighbourhood retail shops for the second group is uncommon in Western developed countries. My opinion!

Line 467

Conclusions

Does the authors find differences among regions?

Author Response

The authors are grateful to reviewer’s for their valuable comments and corrections. We have made revisions to our manuscript based on the feedback provided by the reviewer’s. Our responses are listed below.

The English of the article was reviewed by one of the authors, whose native language is English, and necessary corrections were made.

Reviewer 2

Line

Original version

Suggestions/comments

Responses

Line 14

food safety, food security

I should put the first key issue, “food security”

Revision was made accordingly.

Line 17

To inform

To present?

Revision was made accordingly.

Line 18

Turkiye is in the enviable position…

I would suggest this sentence to the second due to the fact it is the subject of this study.

Revision was made accordingly.

Line 27

Inform dialogue

Please clarify

Revised as “….will increase dialogue between the components of food sector…”

Line 40

From ‘farm to fork’

I would suggest to add more information regarding this frase.

“….including production, processing, distribution and storage.” was added.

Line 41

Data from WHO

I propose to add also the DALY-disability-adjusted life year

“…and 33 million healthy life years (DALYs) are lost…” was added.

Line 45

This is not a hypothesis. Propose introducing the lack of Food Safety Infrastructure in developing countries and the full-scale operation in developed countries.

Sentence was revised accordingly.

Line 47

Shifting demographics

What is the intention of this phrase? Be clearer in explanation.

Sentence was revised for better understanding “….shifting demographics due to refugees….”

Line 63

Food waste

After elaboration of Food safety and food waste key factors, less information is considered about food security in the time of crises.

Added a paragraph regarding food security

Line 66

Please correct. Avoid double spaces.

Deleted.

Line 69

Released into the environment

Please avoid repeatedness

Deleted.

Line 72

Suggest to consider also the transport

Added accordingly.

Line 78

Adversely impacted

Please explain

Sentence revised for better understanding.

Line 91

Study

Please unify the terminology “study” and ”survey”

Terminology was unified.

Line 92

Geographic regions of Turkey

Except the geography, what do you think about the cultural diversity or something else connected with regions?

Türkiye consists of 7 geographical regions. These regions have their own unique culture, life and food habits. Although there is serious human mobility between regions due to working life and education, the survey was performed in 12 provinces to conduct all these 7 regions to ensure that the survey represents the whole country. However, depending on the human population, the number of subjects was kept high in densely populated areas.

Line 111

Please unify the indent

Corrected.

Line 174

A-B; C1-C2, and D-E

Please explain the groups categorization

Defined in the materials and methods section.

Line 179

Quality and taste

Please put in quotes this phrase

Revised accordingly.

Line 180

In my opinion the only option stands ‘ads’ as different compared with others.

Since “ads” are also a factor affecting purchasing preferences, it was asked in the same question with others.

Line 183

Please merge two sentences

Merged accordingly.

Line 212

Today

Please change this word e.g. actually

Changed accordingly.

Line 219

Said foods

Please explain

Unsafe food is meant. Word “said” and previous sentence were revised.

Line 240

Outlets

This word is not so common in food sector. Please unify the terminology according to www.fmi.org/food-industry-glossary

Revised accordingly.

Line 360

Since these three divisions belong to the UK, I would suggest the second. Even though is not stated Scotland.

Revised as “UK”

Line 371

Ofof

Please correct

Corrected.

Line 392

In Turkiye

In my opinion, not only. What about the access to the retail actors? There is a difference in urbanization characteristics between Western countries and developing countries. Quick access to the neighbourhood retail shops for the second group is uncommon in Western developed countries. My opinion!

Thanks for your valuable opinion. Although there is no difference as you mentioned in terms of access to packaged or unpackaged bread in Türkiye, many factors can of course affect this situation. However, since this paragraph focuses only on relation between socio-economic status and packaged bread prefer, it was found more appropriate to make such an inference.

Line 467

Conclusions

Does the authors find differences among regions?

No significant difference was detected regarding the regions where the survey was conducted. The results represent the whole country.

Reviewer 3 Report

Comments and Suggestions for Authors

Dear author/s,

the topic of the manuscript is interesting, however there are some aspects that should be improved:

1. Abstract:  It is not necessary to indicate the subsections of the manuscript. At the same time the analysis method should be mentioned, and the practical implications.

2. Introduction: Changes regarding the consumers' behavior in terms of food waste management during and after Covid-19 pandemic should be presented. I recommend to add a literature review section in which to present these aspects, beside the factors affecting the food waste management, in order to highlight the contribution of the manuscript to the existing background. The research questions should be stated as well, based on the literature review.

3. Material and methods: Please clarified whom applied the face to face survey? How the items used in the questionnaire were evaluated? Were any previous researches used in order to develop the questionnaire? How the consistency and reliability of the questionnaire were tested? The statistics used is quite simple, I recommend more advance statistics. This fact is a must. Could be conducted comparative analyses based on the socio-demographic profile.

4. Results: are clearly presented, but not enough.

5. Discussions: should be organized based on the research questions. In order to emphasize the theoretical and practical implications of the current research.

6. Conclusions: limitations, managerial implications, future research directions should be added.

Good luck!

Author Response

The authors are grateful to reviewer’s for their valuable comments and corrections. We have made revisions to our manuscript based on the feedback provided by the reviewer’s. Our responses are listed below.

The English of the article was reviewed by one of the authors, whose native language is English, and necessary corrections were made.

Reviewer 3

the topic of the manuscript is interesting, however there are some aspects that should be improved:

  1. Abstract: It is not necessary to indicate the subsections of the manuscript. At the same time the analysis method should be mentioned, and the practical implications.

Response: Subsections in the abstract have been prepared in accordance with the journal's writing rules. It has been added to the method section where the results are evaluated statistically.

  1. Introduction:Changes regarding the consumers' behavior in terms of food waste management during and after Covid-19 pandemic should be presented. I recommend to add a literature review section in which to present these aspects, beside the factors affecting the food waste management, in order to highlight the contribution of the manuscript to the existing background. The research questions should be stated as well, based on the literature review.

Responce: Issues related to consumers' waste behavior are stated in the introduction and discussion sections with reference to the relevant literature. Apart from this, the article also focuses on food safety and food security issues. For this reason, all these issues have been tried to be addressed in a balance. It is thought that providing more detailed information would make the article too long.

  1. Material and methods: Please clarified whom applied the face to face survey? How the items used in the questionnaire were evaluated? Were any previous researches used in order to develop the questionnaire? How the consistency and reliability of the questionnaire were tested? The statistics used is quite simple, I recommend more advance statistics. This fact is a must. Could be conducted comparative analyses based on the socio-demographic profile.

Response: Explanation has been added to the relevant section (page 6-7). Questionnaire was developed based on the literature search and the studies conducted on simiar surveys.

  1. Results:are clearly presented, but not enough.

Response: The survey is quite comprehensive and the findings section focuses on sharing meaningful data.

  1. Discussions: should be organized based on the research questions. In order to emphasize the theoretical and practical implications of the current research.

Response: The discussion section has been prepared by discussing the results of the survey on food safety, food security and food waste topics in an order with the available literature information.

  1. Conclusions:limitations, managerial implications, future research directions should be added.

Response: Conclusions relevant with the scope and findings of the study were expressed.

Reviewer 4 Report

Comments and Suggestions for Authors

Please replace the word 'Tűrkiye' with 'Turkey' as 'Tűrkiye' is not an English word. Ensure this correction is made consistently throughout your manuscript. In the Introduction (Lines 32-36), you discuss financial crisis, increasing population, and more, but you overlook an important aspect, which is the ongoing war as indicated by doi: 10.3390/foods11142098. Additionally, in Line 85-87, you mention 'security,' but your title only covers topics such as food safety, purchasing behavior, and food wastage. It is essential to maintain consistency in the focus of your manuscript. You may find the following article of interest: doi: 10.1016/j.spc.2021.03.001. Please provide an overview of the paper's structure. In the 'Material and Methods' section, there are some missing details, such as the distribution method of the survey. Did you employ an online platform like SurveyMonkey? Additionally, please include information about the gender ratio. The 'Results and Conclusion' section is well-written. However, in the 'Conclusion,' you consistently mention 'food security' without presenting any corresponding work or findings. Please remove these references, as they do not align with the content of your manuscript. Finally, please thoroughly review the manuscript for any typos, particularly in Line 371

Comments on the Quality of English Language

Manuscripts needs a thorough check for grammatical errors, typos and sentence formations. 

Author Response

The authors are grateful to reviewer’s for their valuable comments and corrections. We have made revisions to our manuscript based on the feedback provided by the reviewer’s. Our responses are listed below.

The English of the article was reviewed by one of the authors, whose native language is English, and necessary corrections were made.

Reviewer 4

  1. Please replace the word 'Tűrkiye' with 'Turkey' as 'Tűrkiye' is not an English word. Ensure this correction is made consistently throughout your manuscript.

Response: Upon Türkiye’s request, the country's name adopted by the United Nations is Türkiye. For this reason, the term Türkiye is now officially used instead of Turkey.

  1. In the Introduction (Lines 32-36), you discuss financial crisis, increasing population, and more, but you overlook an important aspect, which is the ongoing war as indicated by doi: 10.3390/foods11142098.

Response: Reference and explanation has been added.

  1. Additionally, in Line 85-87, you mention 'security,' but your title only covers topics such as food safety, purchasing behavior, and food wastage. It is essential to maintain consistency in the focus of your manuscript. You may find the following article of interest: doi: 10.1016/j.spc.2021.03.001. Please provide an overview of the paper's structure.

Response: A paragraph on food security was added in line with the suggestions of other referees.

  1. In the 'Material and Methods' section, there are some missing details, such as the distribution method of the survey. Did you employ an online platform like SurveyMonkey? Additionally, please include information about the gender ratio.

Response: Explanation has been added to the relevant section (page 6-7).

  1. The 'Results and Conclusion' section is well-written. However, in the 'Conclusion,' you consistently mention 'food security' without presenting any corresponding work or findings. Please remove these references, as they do not align with the content of your manuscript.

Response: In conclusion section “food security” was deleted.

  1. Finally, please thoroughly review the manuscript for any typos, particularly in Line 371

Response: Revised accordingly.

The English of the article was reviewed by one of the authors, whose native language is English, and necessary corrections were made.

Round 2

Reviewer 1 Report

Comments and Suggestions for Authors

The manuscript has been revised and implemented with the responses to the proposed comments, and they appear thorough. 

Comments on the Quality of English Language

Moderate editing of English language still required

Author Response

The authors are grateful to reviewer’s for their valuable comments and corrections. We think that the article has improved after the criticisms and is now ready to be accepted for publication.

Our responses are listed below.

Reviewer 1: The manuscript has been revised and implemented with the responses to the proposed comments, and they appear thorough.

Moderate editing of English language still required

Response: Thank you. The English of the article was reviewed. One of the authors is a native speaker from England (Dr. Julian Stowell) and he read the manuscript word by word, line by line and conducted necessary corrections.

Reviewer 3 Report

Comments and Suggestions for Authors

Dear author/s,

thank you for the improved version of the manuscript, however there still are some aspects that should be improved:

1. As previous mentioned the literature review part should be revised, in order to underline what gap in the existing literature this research fills.

2. Please clearly present the research instrument. What types of questions, what items and specifically from which previous researches were developed.

3. The results should be reported in e more scientific way, not like in an thesis or report.

4. Please state the limits and future research questions.

other: there are some spelling mistakes. please put the article in the template of the journal.

good luck!

Comments on the Quality of English Language

There are some spelling mistakes.

Author Response

The authors are grateful to reviewer’s for their valuable comments and corrections. We think that the article has improved after the criticisms and is now ready to be accepted for publication.

Our responses are listed below.

Reviewer 3: Thank you for the improved version of the manuscript, however there still are some aspects that should be improved:

  1. As previous mentioned the literature review part should be revised, in order to underline what gap in the existing literature this research fills.

Response: In the article, the gap in the literature about the topic of the article was stated.

. “In this context, in order for food industry stakeholders to become more resilient to crises and create a vision for the future, it is necessary to have realistic information on the main issues affecting consumers' purchasing behaviour. However, in the literature review, it is noted that there are no comprehensive studies across the country in Türkiye, which is an important producer and market in Europe.”

  1. Please clearly present the research instrument. What types of questions, what items and specifically from which previous researches were developed?

Response: Thank you. The survey was conducted face to face. The questions were prepared specifically for the topics to be analysed, taking into account previous studies. Before starting the survey, a pilot survey was carried out by professional interviewers. All the points suggested by the reviewer are stated in the method section.

  1. The results should be reported in e more scientific way, not like in a thesis or report.

Response: First, we would like to clarify that this is not a thesis study, but a study requested by important food industry representatives in Türkiye and carried out by a professional survey company. Due to the abundance of data, it was deemed appropriate to write the findings in this way so that it can be understood more easily by the readers both from scientific community and industry. In addition, the article was evaluated by four valuable reviewers in total and an effort was made to revise it to meet the criticisms of all referees. Thank you in advance for your understanding.

  1. Please state the limits and future research questions.

other: there are some spelling mistakes. please put the article in the template of the journal.

Response:

Conclusion was improved accordingly. “This survey was conducted during the pandemic period, and it is possible that those conditions affected consumer behaviour periodically. For this reason, it is important to repeat these studies at regular intervals so that the food industry can create healthier future visions.”

The English of the article was reviewed. One of the authors is a native speaker from England (Dr. Julian Stowell) and he read the manuscript word by word, line by line and conducted necessary corrections.

Reviewer 4 Report

Comments and Suggestions for Authors

The authors have carried out revision of their manuscript and most of my comments and feedback have been addressed. The manuscript reads well and good enough for publication in this journal.

Comments on the Quality of English Language

Minor editing with regards to usage of commas, typos, and some grammatical errors.

Author Response

The authors are grateful to reviewer’s for their valuable comments and corrections. We think that the article has improved after the criticisms and is now ready to be accepted for publication.

Our responses are listed below.

Reviewer 4

The authors have carried out revision of their manuscript and most of my comments and feedback have been addressed. The manuscript reads well and good enough for publication in this journal.

Minor editing with regards to usage of commas, typos, and some grammatical errors.

Response: Thank you. The English of the article was reviewed. One of the authors is a native speaker from England (Dr. Julian Stowell) and he read the manuscript word by word, line by line and conducted necessary corrections.